# Superhydrophobic Polypyrrole-Coated Cigarette Filters for Effective Oil/Water Separation

**Jialu Zhang [1,2], Hao Xu [2], Jie Guo [2], Tianchi Chen [2,\*] and Hongtao Liu [3,\*]**

[1] College of Electromechanical Engineering, China University of Mining and Technology, Xuzhou 221000, China; zhangjialu0730@gmail.com

[2] College of Electromechanical Engineering, Jiangsu Normal University, Xuzhou 221000, China; xh852152344@gmail.com (H.X.); 3020172111@jsnu.edu.cn (J.G.)

[3] College of Materials and Physics, China University of Mining and Technology, Xuzhou 221000, China

\* Correspondence: 6020180185@jsnu.edu.cn (T.C.); liuht@cumt.edu.cn (H.L.)

**Abstract:** To facilitate the recycling and reuse of cigarette filters and oil/water separation, a superhydrophobic cigarette filter was made by coating with dodecanethiol-modified polypyrrole (Ppy) particles by a dip-coating method. SEM, FTIR, and XPS were used to analyze the surface morphology and chemical compositions. The as-prepared superhydrophobic cigarette filter can realize wettability alteration via changing the ammonium persulfate (APS) concentration from 0.15 mol/L to 3 mol/L, and the contact angle increased from 0° on the original cigarette filter to 155° with a sliding angle of 5°. The superhydrophobic cigarette filter could effectively separate various oils and organic solvents. The separation efficiency was 98.8% and the separation stability was good. Furthermore, the as-prepared superhydrophobic cigarette filter had a large oil absorption range and could absorb different oils and organic solvents, including petroleum ether, engine oil, vegetable oil, n-hexane, and chloroform, with maximum absorption capacities ranging from 9.4 g/g to 22.7 g/g. According to the above results, we believe that the as-prepared superhydrophobic cigarette filter should have great potential in the recovery of solid waste and high-efficiency oil/water separation.

**Keywords:** polypyrrole; superhydrophobicity; oil/water separation; cigarette filter

## 1. Introduction

With the continuous development of industrialization, the demand for oil is also increasing [1,2]. From oil extraction, refining, storage, and transportation to industrial use, these processes inevitably produce a large amount of oil/water mixture waste [3,4]. Accidents such as oil spills and substandard discharge of industrial sewage cause catastrophic damage to marine and lake ecosystems [5]. Thus, effective methods to recycle oil from water are urgently needed.

Many methods have been developed to separate oil and organic contaminants from water, such as gravity separation, centrifugal separation, electric de-separation, coarse-grained dehydration of emulsified water, and air-float separation [6]. However, most of these methods are inefficient, poor-safety, high-energy-consumption, and complicated processes [7], making them challenging for use in large-scale applications. Besides the traditional methods, superhydrophobic and oleophilic materials, which can repel water and be wetted by oils, have been considered as suitable candidates for high-efficiency oil/water separation. Recently, researchers have fabricated an enormous amount of superhydrophobic materials for oil/water separation through sol–gel [8], chemical etching [9], spray methods [10], and hydrothermal treatment [11]. Superhydrophobic materials can be divided into separation materials and absorption materials. Separation materials mainly use steel mesh, textiles, and kapok fiber as substrates [12,13], which simply separate oil and water. Raju et al. [14] reviewed the recent

progress of oil/water separation technologies based on filtration and absorption methods using various materials that possess surface superwetting properties. Gary et al. [15] prepared a pair of stainless steel meshes with hydrophilic and hydrophobic functions to achieve rapid and continuous separation of oil/water mixtures. Chen et al. [16] utilized laser marking technology to fabricate superhydrophobic stainless steel mesh for oil/water separation. Zhang et al. [17] utilized the reverse-phase method initiated by an inert solvent to fabricate a superhydrophobic PVDF membrane for high-flux oil/water emulsion separation. Lee et al. [18] utilized the electrostatic spinning deposition method to fabricate a superhydrophobic membrane for selective oil/water separation. The main problems for oil/water separation materials include their absorption capacity, secondary pollution, and separation stability, which limit their industrial application [19]. Absorption materials make use of 3d porous materials as substrates, such as sponges and foams. Oil/water separation is one of the functions of adsorption materials, but another important function is that they can absorb oil from water. The absorbed oil can be easily reused by squeezing or burning, which avoids secondary pollution [20]. Li et al. [7] fabricated superhydrophobic $Fe_3O_4$ melamine sponges for absorbing oil from water. The modified sponges have high absorption capacities and good reusability. Qiao et al. [21] fabricated graphene-based superhydrophobic sponges for oil/water separation, which showed excellent absorption properties for various oils and organic contaminants. Jin et al. [22] fabricated a joule-heated graphene-wrapped sponge for viscous oil fast sorption. Wang et al. [23] fabricated a micro/nanostructure layer sponge which can be recycled for more than 70 cycles.

Compared to these high-porosity and expensive sponges, cigarette filters can be an economical choice for fabricating superhydrophobic absorbing materials. Cigarette filters are the most troublesome solid waste in the world [24]. Their main constituent material is cellulose acetate, which cannot easily biodegrade [25]. If cigarette filters can be used as an oil/water separation material, this can effectively solve the problem of the difficult recycling and degradation of waste cigarette filters. In the research process, we found that there are few cases of oil/water separation in the field of cigarette filter recycling [26–28]. The polymerization of silane coupling agents is the main method used in the literature to prepare superhydrophobic cigarette filters. Herein, we prepared superhydrophobic cigarette filters via polypyrrole formation and n-dodecanethiol modification. The polypyrrole particles were successfully deposited on the fiber surface, which produced hierarchical micro/nanostructures to realize superhydrophobicity. The synthesis and deposition of the polypyrrole particles was achieved by a dip-coating method, which was relatively facile and cheap compared with the methods previously reported for superhydrophobic cigarette filters. Moreover, the as-prepared superhydrophobic cigarette filters exhibited excellent water repellency with a water contact water of 155°, highly efficient oil separation ability, separation stability, and great absorption capacity; the superhydrophobic Ppy-coated cigarette filter absorbed oils and organic solvents with absorption capacities 9.4–22.7 times the filter's mass. This result shows that the superhydrophobic cigarette filters can not only effectively separate oil/water mixtures but also promote waste recycling and environmental protection.

## 2. Experimental Sections

### 2.1. Materials

Cigarette filters were collected from China University of Mining and Technology. Ammonium persulfate, and pyrrole (Py) were purchased from Sinopharm Chemical Reagent Co., Ltd, Shanghai, China. N-hexane, chloroform, n-dodecanethiol and petroleum ether were purchased from Aladdin Company, Shanghai, China. The vegetable oil was purchased from Singapore Guo Brothers Grain and Oil Pte Ltd., Singapore. The engine oil was purchased from Royal Dutch /Shell Group of Companies, Zhejiang, China.

## 2.2. Preparation of Superhydrophobic Cigarette Filters

First, cigarette filters with a length of 3 cm and a diameter of 0.8 cm were ultrasonically cleaned in ethanol for 20 min to remove impurities and then dried at 60 °C for 30 min. A quantity of 0.2–2 mol pyrrole was added into 10 mL ethanol. The solution was stirred for 10 min. The cigarette filters were soaked in the pyrrole ethanol solution for 5 min. A quantity of 0.3–3 mol ammonium persulfate was added into 10 mL deionized water and stirred for 5 min. After removal from the pyrrole solution, cigarette filters were directly immersed in the ammonium persulfate solution for 1 min. Then, reacted cigarette filters were rinsed with deionized water to remove the unreacted reagent and solvent and placed in a dry oven at 60 °C for 1.5 h. Finally, Ppy cigarette filters were immersed in an ethanol solution of n-dodecanethiol at a concentration of 2.5 wt.% for 5 h, and then placed in an oven at 60 °C for 1.5 h.

## 2.3. Characteristics of the Superhydrophobic Cigarette Filters

The surface morphologies of the Ppy cigarette filters were analyzed by Quanta 400 FEG SEM (FEI Company, Hillsboro, OR, USA). Functional groups on the cigarette filters' surfaces were detected by FTIR VERTEX 80v (Bruker, Germany). The surface composition and elemental valence of Ppy particles on the cigarette filters' surfaces were determined by XPS ESCALAB 250Xi (Thermo Fisher Scientific, Waltham, MA, USA). The water contact angle (WCA) and sliding angle (WSA) were measured using a JC2000 contact angle meter (Zhongchen, China). A 5 μL deionized water droplet was used to measure the water contact angle on the sample surface at room temperature.

## 2.4. Measurements of Oil Separation and Absorption

A simple gravity oil/water separation method was used to measure the oil separation efficiency. It consisted of a funnel and cigarette filters. Cigarette filters were pulled together with a steel string. These wired cigarette filters were squeezed into the neck of funnel. We poured the oil/water mixture into the funnel and waited for gradual separation. The oil/water separation efficiency ($Q_{se}$) was calculated via Equation (1):

$$Q_{se} = \frac{(m_1 - m_2)}{m_0} \times 100\%. \tag{1}$$

In Equation (1), $m_1$ and $m_2$ respectively represent the weights of the oil/water mixture before and after separation, and $m_0$ represents the initial weight of oil.

In order to evaluate the oil absorption properties of the superhydrophobic cigarette filters for different oils and organic solvents, our experiment method for oil and organic solvent absorption comprised the following steps: an as-prepared cigarette filter was immersed in the oil, then the cigarette filter was taken out when it reached saturation. The absorption capacity ($Q_{ab}$) was calculated according to the following relation in Equation (2):

$$Q_{ab} = \frac{(m_1 - m_0)}{m_0} \ \text{g/g} \tag{2}$$

where $m_0$ and $m_1$ represent the weights of the cigarette filter before and after adsorption, respectively.

## 3. Results and Discussion

### 3.1. Surface Morphology and Characterization of Superhydrophobic Cigarette Filters

A SEM image of the original cigarette filter surface is shown in Figure 1a. The experimental sample revealed porous fiber structures, which benefited oil absorption. The fiber surface presented smooth and flat topography, as shown in the inset of Figure 1a. Figure 1b shows images of a Ppy-coated cigarette filter surface with 0.2 mol/L pyrrole under low and high magnification. After the polymerization of pyrrole, the fiber surface became rough. Ppy nanoparticles were spherical in structure and evenly

distributed, and the sizes of the particles were about 400 nm. The nanoparticles did not completely cover the fiber surface. Moreover, some Ppy nanoparticles aggregated into microparticles, as shown in the inset of Figure 1b. With increasing pyrrole concentration, the surface became increasingly rougher with micro–nano hierarchical structures, as shown in Figure 1c. The Ppy particles were in an obvious aggregation state, and the average particle size increased to 1 μm. Thus, Ppy nanoparticles uniformly covered the cigarette filter surface, and the roughness of the fiber surface could be easily controlled by the pyrrole concentration.

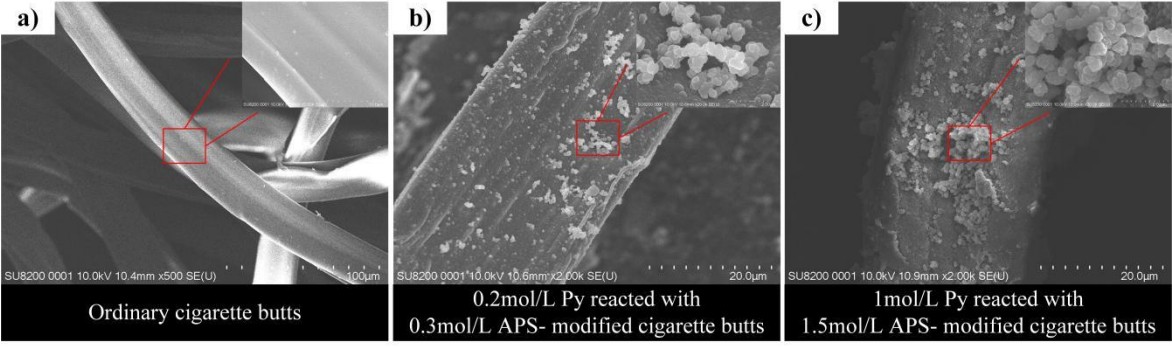

**Figure 1.** SEM images of Ppy-coated cigarette filters with different pyrrole concentrations: (**a**) Ordinary sample; (**b**) 0.2 mol/L pyrrole; (**c**) 1.0 mol/L pyrrole.

The FTIR spectrum for a superhydrophobic cigarette filter is shown in Figure 2. The characteristic peaks at 2921 cm$^{-1}$ and 2851 cm$^{-1}$ were attributed to C−H stretching vibration of −CH$_3$ and −CH$_2$−. Moreover, there were no obvious S-H characteristic peaks at 2570 cm$^{-1}$. Combined with the existence of −CH$_3$ and −CH$_2$−, this confirmed that S−H is broken, and n-dodecanethiol is grafted on the Ppy particle surface via chemical interaction [29]. The C=C stretching vibration was located at 1550 cm$^{-1}$. The deformation vibration absorption peak of C–H was located at 1463 cm$^{-1}$. The C–N stretching vibration peaks were located at 1200 cm$^{-1}$ and 1037 cm$^{-1}$. The =C–N plane vibration was located at 898 cm$^{-1}$ [30]. A stretching vibration peak of C=O was located at 1717 cm$^{-1}$, owing to powder oxidization in the air [31]. Consequently, it could be concluded that the cigarette filter was covered by Ppy particles, and n-dodecanethiol was successfully modified on the surface of Ppy particles.

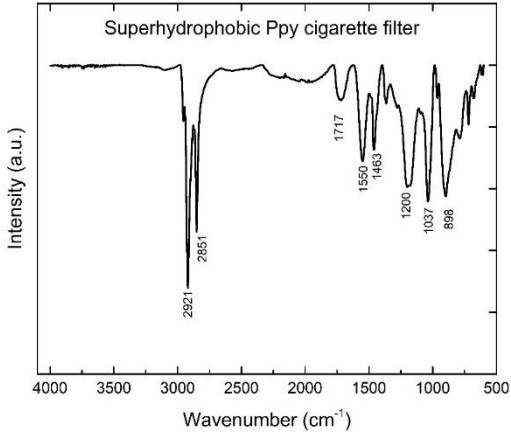

**Figure 2.** FTIR spectrum of an as-prepared superhydrophobic cigarette filter.

XPS was used to investigate the surface chemical composition and valence state. In order to further analyze the polymerization mechanism of pyrrole and the reaction mechanism of n-dodecanethiol grafting, typical XPS survey spectra of Ppy particles before and after modification with n-dodecanethiol are shown in Figure 3a. Both samples exhibited a C1s peak at 284.8 eV, O1s peak at 531.9 eV, and N1s peak at 399.9 eV. After n-dodecanethiol modification, two new peaks appeared in the spectrum of Ppy particles at 163.84 eV and 197.84 eV, which were the binding energy peaks of S2p and S2s, respectively. As shown in Figure 3b, the C1s spectrum of Ppy particles showed four different peaks at 283.85 eV, 284.8 eV, 286.0 eV, and 287.3 eV, which corresponded to the characteristic peaks of C-H, C-C, C=C, and C-N, respectively [32]. Figure 3c shows the N1s spectrum of Ppy particles. The characteristic peaks at 399.5 eV, 399.8 eV, and 401.0 eV corresponded to C-N, N-H, and $NH_4^+$ groups, respectively [33]. The characteristic peaks of C1s and N1s in the XPS spectra were highly consistent with the FTIR spectrum, which further proved that Ppy particles were indeed generated on the cigarette filter surface. Figure 3d shows the O1s spectrum of Ppy particles. The XPS O1s peaks could be fitted to two peaks, located at 532.1 and 534.9 eV, which were attributed to the hydroxyl and surface-adsorbed oxygen species [34]. The S2p XPS spectrum of the Ppy particles modified by dodecanethiol is shown in Figure 3e. It could be fitted to three major peaks. The peak located at 162.8 eV was attributed to C-S bonds between sulfur of n-dodecanethiol and Ppy nanoparticles [35]. The peak located at 163.3 eV was attributed to unbound thiols [36]. The S2p peak at 164.4 eV was attributed to S–R, which gives Ppy particles hydrophobicity [37]. The C1s spectrum of Ppy particles modified by dodecanethiol is shown in Figure 3f. It could be fitted to five major peaks. The peaks located at 283.8 eV, 284.8 eV, 285.2 eV, 286.3 eV, and 297.5 eV were attributed to C-H, C-C, C-S, C=C, and C-N, respectively [38]. The marked appearance of the C-S peak component and the appearance of the S2p signal peak indicated that dodecanethiol was successfully grafted onto the Ppy filter fiber surface.

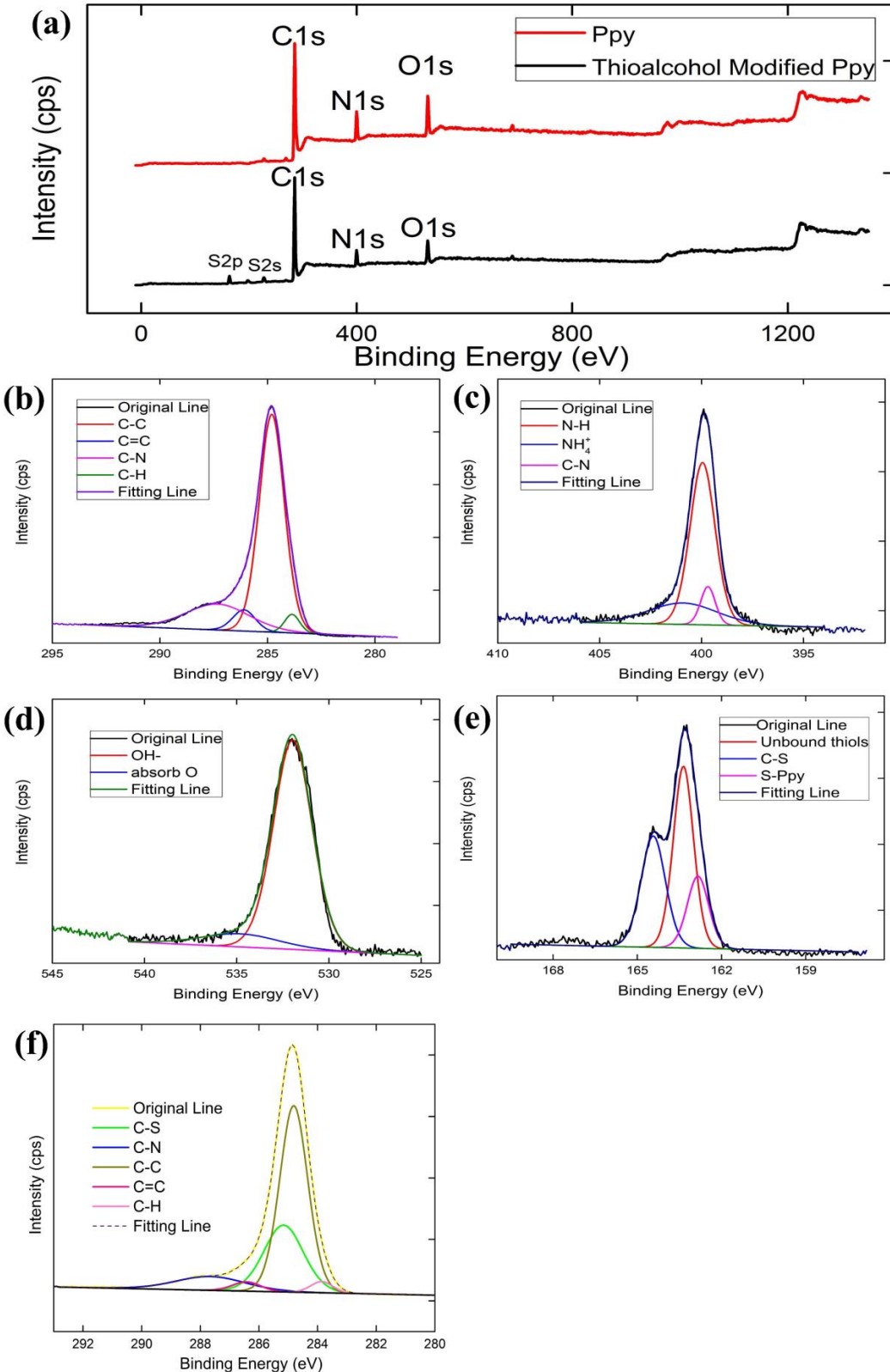

**Figure 3.** XPS survey spectra of Ppy particles before and after modification with n-dodecanethiol: (**a**) Wide survey spectra; (**b**) C1s spectra of the original cigarette filter; (**c**) N1s spectra of the original cigarette filter; (**d**) O1s spectra of the original cigarette filter; (**e**) S2p spectra of the n-dodecanethiol-modified Ppy-coated cigarette filter, and (**f**) C1s spectra of the n-dodecanethiol-modified Ppy-coated cigarette filter.

## 3.2. Surface Wettability

Figure 4a shows photographs of water and oil droplets on the surface of an ordinary cigarette filter and a Ppy cigarette filter modified by dodecanethiol. A significant improvement in the wettability of the cigarette filter was observed. When the water droplet or oil droplet was dribbled on the original cigarette filter surface, the original cigarette filter absorbed both water droplets and oil droplets. Water droplets and oil droplets were completely spread on the original cigarette filter surface because of the inherently superhydrophilicity and superoleophilicity. Different from those on the unmodified cigarette filter, water droplets remained in a spherical shape on the Ppy cigarette filter surface modified by dodecanethiol, and oil droplets quickly penetrated into the interspace between the fibers of the Ppy cigarette filter modified by dodecanethiol. The results illustrated that the Ppy cigarette filter modified by dodecanethiol has superhydrophobicity and superlipophilicity. Figure 4b shows contact angle images of the modified filters with different pyrrole contents. When the pyrrole content was 0.1 mol/L, the water droplet remained hemi-ellipsoidal, indicating that the modified filter surface was hydrophobic. When the pyrrole content reached 1 mol/L, the water droplet had a spherical shape, which means that the filter surface was superhydrophobic.

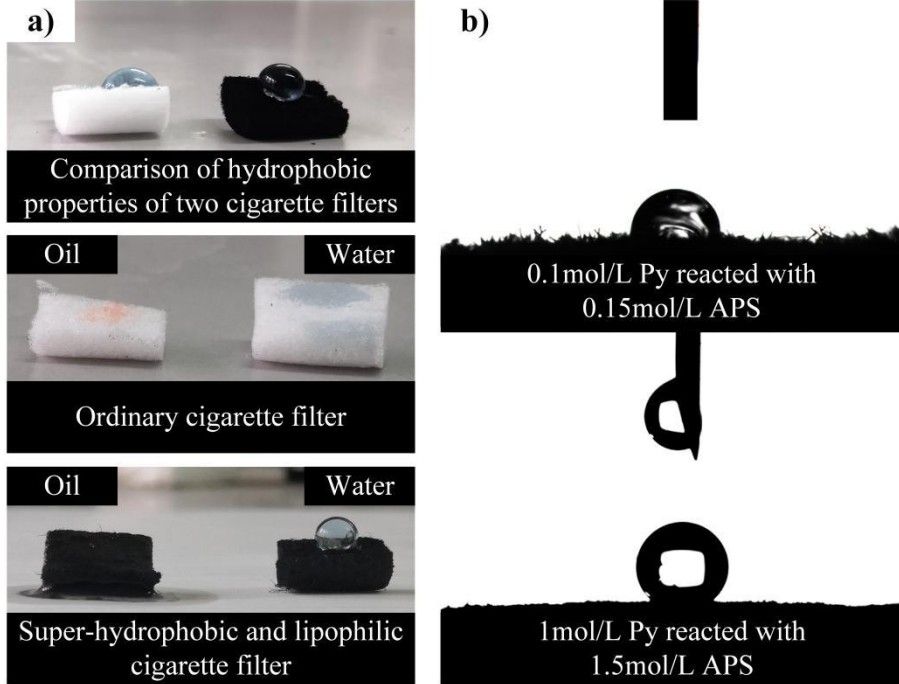

**Figure 4.** (**a**) Photographs of the wettability of an ordinary cigarette filter and a dodecanethiol-modified Ppy cigarette filter; (**b**) Contact angle images of dodecanethiol-modified Ppy cigarette filters with different Py concentrations.

In order to further investigate the relationship between pyrrole content and surface wettability, the wettability of the modified filter was tested for different pyrrole contents, as shown in Figure 5. The contact angle increased with increasing pyrrole content, while the sliding angle decreased. When the pyrrole content was 0.5 mol/L, the contact angle was 150°, and the sliding angle was about 10°, fulfilling the requirements of superhydrophobicity. According to our SEM images, the pyrrole concentration can change the surface morphology of the filter. To our knowledge, the transition of the wettability of the cigarette filter from superhydrophilic to superhydrophobic can be realized by changes in the surface morphology and surface energy [39]. When the pyrrole content was lower, the filter surface was covered with unevenly distributed tiny Ppy nanoparticles. The water could wet the nanostructures but not spread on the filter surface owing to the hydrophobicity of dodecanethiol, which could be

explained by the Wenzel mode [40]. With increasing pyrrole concentration, a large amount of Ppy particles aggregated and formed micro–nano hierarchical structures on the filter surface. With the help of low-surface-energy material, there was solid–liquid–vapor composite contact between the micro–nano hierarchical structures and water, which was consistent with the Cassie model [41]. Thus, a water droplet could easily roll off the modified cigarette filter surface owing to the existence of the air layer.

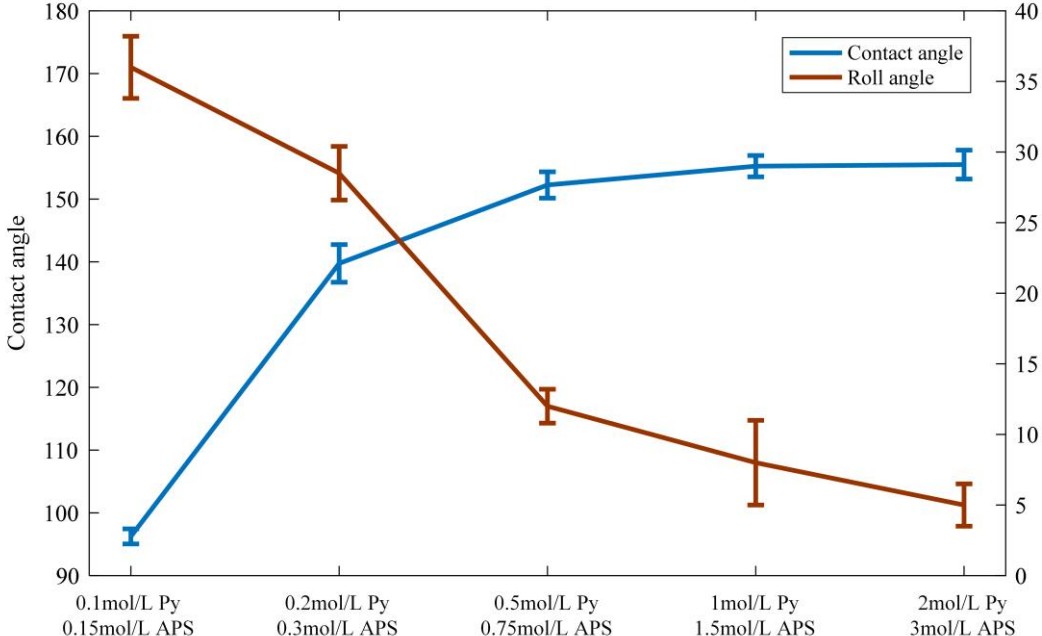

**Figure 5.** Pyrrole content versus surface wettability.

### 3.3. Oil Separation Ability

Figure 6 shows the processes of oil/water separation for the original cigarette filter, Ppy-coated cigarette filter, and superhydrophobic Ppy-coated cigarette filter. For the original cigarette filter, when a mixture solution of petroleum ether (dyed with Sudan IV) and water (dyed with blue ink) was poured in the funnel, the cigarette filter firstly absorbed petroleum ether. The petroleum ether dripped into the measuring cylinder. When the water came into contact with the cigarette filter, the water, replacing the petroleum ether, started to drip into the measuring cylinder owing to the difference in density. Thus, the original cigarette filter failed to separate oil and water, as shown in Figure 6a. For the superhydrophobic Ppy-coated cigarette filter, when the mixture solution was poured in the funnel, the cigarette filter quickly absorbed petroleum ether, and the petroleum ether continuously dripped into the measuring cylinder. Owing to the water repellent, water could not penetrate into the pores of the modified cigarette filter. Thus, water and petroleum ether were separated efficiently by the modified cigarette filter, as shown in Figure 6b.

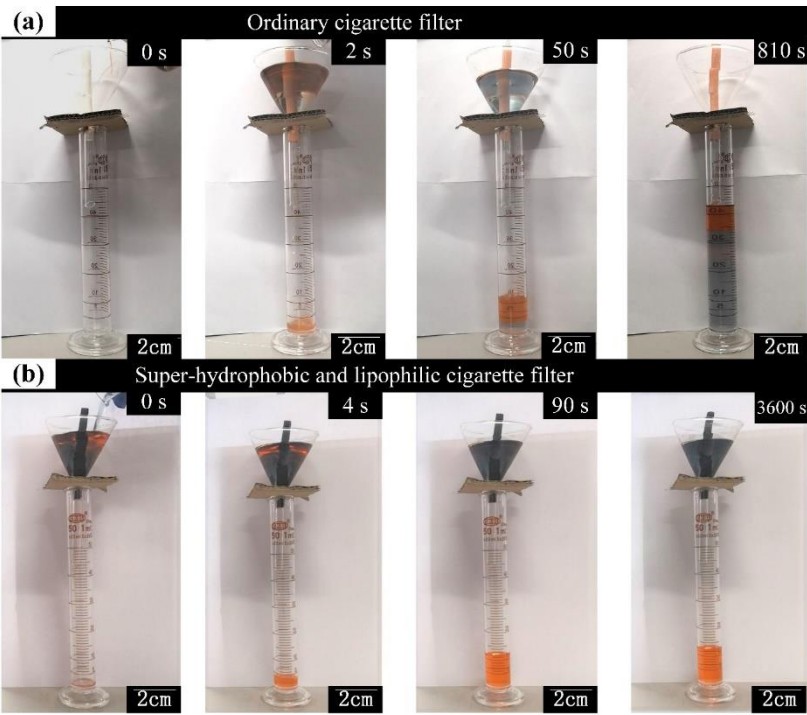

**Figure 6.** Photos of oil/water separation: (**a**) ordinary sample; (**b**) superhydrophobic sample.

Figure 7a shows the efficiency of the modified cigarette filter for different oil/water separation cases. Both light and heavy oil could be separated by the modified cigarette filter. For light oil, the separation efficiency of the modified cigarette filter was 98.8% for an engine oil/water mixture. For heavy oil, like chloroform, the modified cigarette filter could separate greater than 98.5% by weight of water from the chloroform/water solution. Figure 7b shows the relationship between the separation efficiency and the number of cycles. Separation efficiency showed a downward trend. After 30 separation cycles of different oil/water mixture solutions, the separation efficiency for the petroleum ether/water mixture still reached 97.5%. Moreover, the separation efficiency reached 96.5%, 96.3%, and 97.3% for an engine oil/water mixture, vegetable oil/water mixture, and chloroform/water mixture, respectively, which revealed the good separation stability of superhydrophobic Ppy-coated cigarette filters for different separation cases of oil/water mixture solutions.

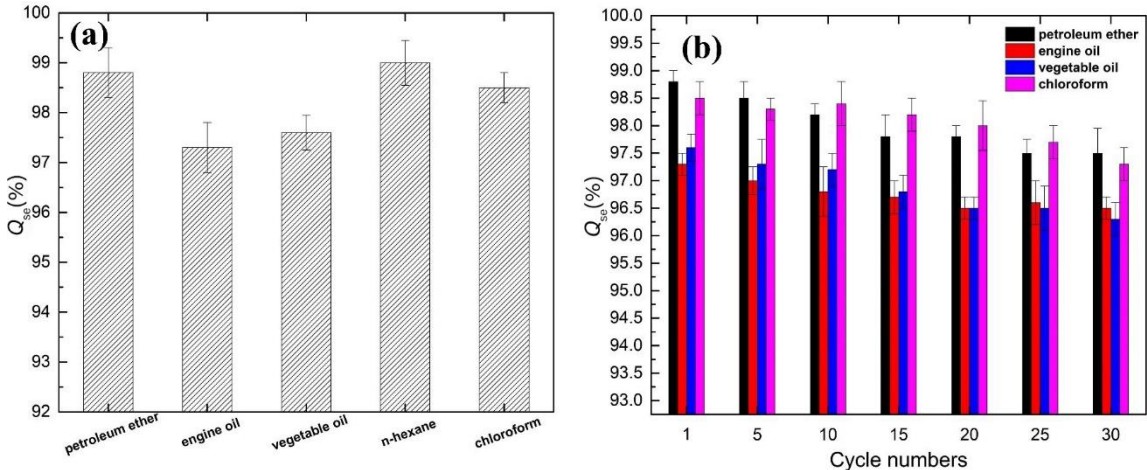

**Figure 7.** Plots of separation efficiency and stability: (**a**) $Q_{se}$ for different oils and organic solvents; (**b**) $Q_{se}$ for different cycle numbers.

### 3.4. Oil Absorption Capacity

Figure 8a shows the oil absorption capacity of different superhydrophobic cigarette filter samples. The 3D porous fiber structures could absorb large amounts of oil [42]. The original cigarette filter and filters prepared using 0.1 mol/L Py, 0.2 mol/L Py, 0.5 mol/L Py, 1 mol/L Py, and 2 mol/L Py had absorption capacities for petroleum ether of 4.6 g/g, 5.4 g/g, 6.8 g/g, 6.95 g/g, 8.1 g/g, and 9.4 g/g, respectively. The absorption capacity of the as-prepared cigarette filter was larger than that of the original cigarette filter for petroleum ether. As the pyrrole concentration increased, the oil absorption capacity increased. The filter prepared using 2 mol/L Py had the largest capacity for petroleum ether. We introduced this indicator to illustrate that the filters can absorb oils up to many times their own mass. The comparison between before and after treatment indicated that the filter could absorb more oil after treatment. Figure 8b shows the oil absorption capacity for different oils and organic solvents; in these experiments, we used filters prepared using 2 mol/L Py. The superhydrophobic cigarette filter had different absorption capacities for different oils and organic solvents because of their physical and chemical properties [43]. The $Q_{ab}$ values for petroleum ether, engine oil, vegetable oil, n-hexane, and chloroform were 9.4 g/g, 13.6 g/g, 13.9 g/g, 9.6 g/g, and 22.7 g/g, respectively. Thus, the high roughness provided by Ppy particles obviously enhanced the filters' absorption capacity for oils or organic solvents. Herein, the filter we made has a large oil absorption range and the ability to adsorb a variety of oils.

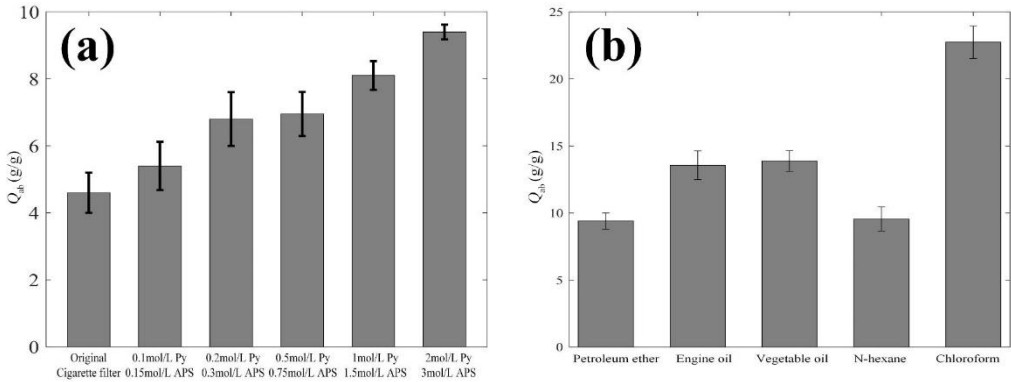

**Figure 8.** The absorption capacities of the as-prepared superhydrophobic cigarette filters: (**a**) $Q_{ab}$ values for different Py concentrations; (**b**) $Q_{ab}$ values of filters prepared using 2 mol/L Py for various oils and organic solvents.

## 4. Conclusions

In this study, we demonstrated a simple method to prepare cigarette filters capable of oil separation and absorption. They are made of waste cigarettes and can solve issues relating to the recycling and reuse of cigarette filters and oil/water separation. Ppy was coated on the surface of the cigarette filter skeleton to achieve micro–nano hierarchical structures. The as-prepared superhydrophobic cigarette filters can realize wettability alteration via changing the ammonium persulfate (APS) concentration from 0.15 mol/L to 3 mol/L; the contact angle increased from 0° on the original cigarette filter to 155° with a sliding angle of 5°. N-dodecanethiol was used as a low-surface-energy material to increase the hydrophobicity. The obtained Ppy-coated cigarette filter modified by N-dodecanethiol exhibited excellent superhydrophobicity and superoleophilicity. In addition, the pyrrole concentration affected the surface wettability. Owing to these outstanding properties, the superhydrophobic Ppy-coated cigarette filter showed very high efficiency in oil/water separation for various oil/water mixtures under gravity alone. The separation efficiency reached 97.3%–99% for both light oil and heavy oil. The superhydrophobic Ppy-coated cigarette filter also had good separation stability. Moreover, the superhydrophobic Ppy-coated cigarette filter absorbed oils and organic solvents with absorption capacities 9.4–22.7 times the filter's mass, which indicated good oil absorption capacities. Therefore,

this superhydrophobic Ppy-coated cigarette filter is an excellent candidate for oil/water separation and absorption; it could not only reduce solid waste but also be applied to clean up oil spills.

**Author Contributions:** J.Z. designed the fabrication process of the superhydrophobic cigarette filters with H.X., and conducted the research with H.X. and J.G., J.Z. finally wrote the manuscript; T.C. rechecked the manuscript; H.L. provided the experiment location and equipment. All authors have read and agreed to the published version of the manuscript.

**Funding:** This study was funded by the Laboratory Society Foundation of Jiangsu Higher Education Institutions of China (No. GS2019ZD07), Jiangsu Normal University Undergraduate Education Research Foundation (No. JYKTY201905), China National Natural Science Foundation (No. 51775545), and Jiangsu Higher Education Institutions Science Foundation of China (No. 19KJD460003).

**Acknowledgments:** The experimental process was guided by experts from the Xuzhou Engineering Machinery Design and Manufacturing Engineering Technology Research Center. We are very grateful for their help.

**Conflicts of Interest:** This article is the original work of the authors and does not involve any conflict of interest.

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
