# Peer review of "Superhydrophobic Polypyrrole-Coated Cigarette Filters for Effective Oil/Water Separation"

_applsci, doi:10.3390/app10061985_

Round 1

Reviewer 1 Report

The authors report interesting work on the modification of cigarette filters to be used in oil-water separation. There are a few points which are of serious concern.

              The attachment of dodecanethiol seems strange. The authors should add more proof that the surface modification is actually occurring. Thiols have been widely reported for the surface modification of metals particularly gold, but a I have never heard of polypyrrole being modified in this was. The FTIR and XPS do not seem to prove that the modification was successful. The formation of an –S-O- bond seems very unusual.

              Figure 9b is very misleading. It suggests that chloroform is much more highly adsorbed than hexane. But the density of the liquids is not accounted for? Chloroform is 3x more dense than hexane.

              The introduction is well written but focuses too much on different surface modification techniques (13-20). It should include more references to oil-water separation reviews such as Journal of Materials Chemistry A 5 (31), 16025-16058.

              Are the cigarette filters used or unused, i.e. has tobacco smoke been passed through them or not? Surely recycling unused filters is a waste of resources, whereas if they have been used before can they be modified in the same way?

              Originally, the cigarette filters are hydrophilic with a contact angle of 0°. This means that they would make excellent water selective filters without further modification. So why do the authors modify the filters at all? There is no need. Alternatively, a combination of modified and unmodified filters would make an excellent combination of materials to use. The authors should read Macromolecular Materials and Engineering 301 (9), 1032-1036 to see that making the surface hydrophobic is not required for successful oil-water separation.

The amount of oil which can be absorbed by the filters is extremely limited. A much better system is one such as ACS applied materials & interfaces 7 (34), 18915-18919, where thousands of liters of liquid can be purified quickly. Again, this reference should be added to the introduction.

Reviewer 2 Report

The authors provide data which are scarce in the literatura thus the novelty and importance of this work. This paper is a valuable contribution and I recommend the publication after minor corrections/suggestion are given below

  1. Authors should improve the introduction and the abstract. They should show the work to be done in a clearer way.
  2. Line 90: Ppy means pyrrole particles? this term is not defined along the text, it should be included
  3. The supplier of hexane, chloroforme, engine civil and vegetable oil used in this work should be included (Section 2.1. Materials)
  4. The authors did a good job but this is not reflected in the article. They should better explain the development of the work, all the studied variables together with the obtained results, so that they can reflect the work performed.
  5. The Figure 9 should be better explained.
  6. What is the superhydrophobic cigarette filter used in Figure 9b?
  7. The conclusión should be improved.

Reviewer 3 Report

in my opinion the paper "Superhydrophobic Polypyrrole Coated Cigarette Filters for Effective Oil/Water Separation" by Zhang et al can be published on Applied Sciences journal. The paper takes into consideration an interesting aspect concerning the recycling of a very common and widespread waste: cigarette filters.

Page 2 line67: choose between fabricated and prepared

Line 79 new filters have been used?

Line 86: explain the role of APS in the reaction, has FeCl3 also been added?

Line 142 figure 2: it might be interesting to show the spectra of cellulose acetate, polypyrrole and dodecanthiol

Line 191: reveals hydrophobicity instead of revealshydrophobicity

Line207: reference for the Wenzel model

Line 213 (figure 6): add legend on the ordinate (Roll angle)

Line 241 (figure 8): add Qse to the ordinate instead of Separation efficiency (%)

Line 242: add (Qse)

Line 248: g/g means g of oil/g of cigarette filter?

Line 258: add Qab to the ordinate instead of oil adsorption ratio

Lin 259 add (Qab)

How do the authors intend to treat used filters from solid urban waste? will they be washed? shredded? is it possible to imagine a process scheme?

Round 2

Reviewer 1 Report

The authors have successfully addressed the majority of my comments, the paper is now ready for publication.